# The Validity of an Updated Metabolic Power Algorithm Based upon di Prampero’s Theoretical Model in Elite Soccer Players

**DOI:** 10.3390/ijerph17249554

**Published:** 2020-12-20

**Authors:** Cristian Savoia, Johnny Padulo, Roberto Colli, Emanuele Marra, Allistair McRobert, Neil Chester, Vito Azzone, Samuel A. Pullinger, Dominic A. Doran

**Affiliations:** 1Research Institute for Sport and Exercise Sciences (RISES), Tom Reilly Building, Liverpool John Moores University, Liverpool L2 2ER, UK; cristiansavoia@gmail.com (C.S.); a.p.mcrobert@ljmu.ac.uk (A.M.); n.chester@ljmu.ac.uk (N.C.); pullinger.s@hotmail.com (S.A.P.); D.A.Doran@ljmu.ac.uk (D.A.D.); 2Department of Biomedical Sciences for Health, Università degli Studi di Milano, 20122 Milan, Italy; 3Department of Biotechnological and Applied Clinical Sciences, University of L’Aquila, 67100 L’Aquila, Italy; colliroberto1954@libero.it; 4School of Sport Sciences and Exercise, Faculty of Medicine and Surgery, Tor Vergata University, 00133 Rome, Italy; emanumarra@libero.it; 5Federazione Italiana Giuoco Calcio (F.I.G.C.), 00198 Rome, Italy; vitoazzone79@gmail.com

**Keywords:** energy cost, soccer-specific circuit, global position system, intermittent exercise, portable gas analyzer

## Abstract

The aim of this study was to update the metabolic power (MP) algorithm ($P_{\dot{V}O_{2}}$, W·kg^−1^) related to the kinematics data (P_GPS_, W·kg^−1^) in a soccer-specific performance model. For this aim, seventeen professional (Serie A) male soccer players ($\dot{V}O_{2\max}$ 55.7 ± 3.4 mL·min^−1^·kg^−1^) performed a 6 min run at 10.29 km·h^−1^ to determine linear-running energy cost (C_r_). On a separate day, thirteen also performed an 8 min soccer-specific intermittent exercise protocol. For both procedures, a portable Cosmed K4b^2^ gas-analyzer and GPS (10 Hz) was used to assess the energy cost above resting (C). From this aim, the MP was estimated through a newly derived C equation (P_GPSn_) and compared with both the commonly used (P_GPSo_) equation and direct measurement ($P_{\dot{V}O_{2}}$). Both P_GPSn_ and P_GPSo_ correlated with $P_{\dot{V}O_{2}}$ (r = 0.66, *p* < 0.05). Estimates of fixed bias were negligible (P_GPSn_ = −0.80 W·kg^−1^ and P_GPSo_ = −1.59 W·kg^−1^), and the bounds of the 95% CIs show that they were not statistically significant from 0. Proportional bias estimates were negligible (absolute differences from one being 0.03 W·kg^−1^ for P_GPSn_ and 0.01 W·kg^−1^ for P_GPSo_) and not statistically significant as both 95% CIs span 1. All variables were distributed around the line of unity and resulted in an under- or overestimation of P_GPSn_, while P_GPSo_ routinely underestimated MP across ranges. Repeated-measures ANOVA showed differences over MP conditions (*F*_1,38_ = 16.929 and *p* < 0.001). Following Bonferroni post hoc test significant differences regarding the MP between P_GPSo_ and $P_{\dot{V}O_{2}}$/P_GPSn_ (*p* < 0.001) were established, while no differences were found between $P_{\dot{V}O_{2}}$ and P_GPSn_ (*p* = 0.853). The new approach showed it can help the coaches and the soccer trainers to better monitor external training load during the training seasons.

## 1. Introduction

The application of new technologies has resulted in the evolution of performance analysis providing appropriate means to track, capture and analyse movement characteristics of soccer players [1]. Many performance variables are routinely captured during match-play/training, helping determine player activity and assess individual player performance profiles to design personalised and novel training approaches [2,3,4]. Match-analysis observations have extensively reported that soccer players typically cover 9–14 km during a game with high-intensity running between 5 and 15%, and running speeds are used as the main parameter for the classification of soccer activity [5,6,7]. Moreover, focusing on running speeds, which comprise of continual changes in speed and direction, is not deemed appropriate for soccer [8,9], and it has therefore been suggested that using “critical” metabolic power (MP) derived from a variable-speed activity is more suitable [10,11]. Thus, attributing a specific metabolic demand related to acceleration and deceleration actions, such as expanding on the description of gameplay with an energetic approach, appears most useful for this purpose [12,13,14].

However, the metabolic demand imposed by soccer match-play and training estimates is calculated from energy cost paradigms derived from laboratory models of constant-speed linear running that do not reflect the totality of soccer-related actions [13]. To effectively measure the energy cost of constant-speed linear-running (C_r_) in soccer_,_ several factors influence the accuracy of energy cost (C) determination [15]. First, the original algorithms were based on running on a compact terrain (e.g., a treadmill; [16,17]). Second, running on a grass surface and also the appropriateness of the footwear has been shown to elevate C_r_ by ~30% and impact kinematics, respectively, when compared to running on a dense terrain [18]. Third, fitness levels have been shown to influence C_r_ as running economy can easily be improved through training [19,20], with research on professional soccer players showing a higher C_r_ by 14% from pre-season compared to in-season [21].

Osgnach et al. [13] based their MP equation on the research conducted by Minetti et al. [17] and added the Pinnington and Dawson [18] correction, which consists of a multiplication term of 1.29 (KT = 1.29) to reflect the difference between running on a treadmill and grass. However, the incorporation of this multiplication term may impact the accurate determination of the C of soccer-specific activities as the KT was developed in recreational runners, which may not represent a suitable kinematic model of running in elite soccer [18]. Therefore, the multiplication coefficient needs to be revisited to ensure a better and more accurate representation of the metabolic constant expressed in elite soccer players on soccer-specific activities [22]. As aforementioned, Osgnach et al.’s [13] investigation does not fully reflect the accurate C_r_ imposed by running on grass and did not use elite athletes, therefore potentially underestimating MP.

More recently, Buchheit and Simpson [23] discussed the efficacy of the MP approach considering the difference between the direct measurement of metabolic power ($P_{\dot{V}O_{2}}$) and the indirect approach proposed by Osgnach et al. [13]. It was observed that several limitations contribute to the underestimation of metabolic power through the global positioning systems (P_GPS_) relating to the GPS sampling frequency about the mathematical model data [24] and the soccer simulation protocol used for the MP calculation [25]. In addition, soccer-specific work–rest: ratios [26] and movements-activities [27] (e.g., sprint, running, jogging, walking, etc.) need to be incorporated. Therefore, additional research is needed to clarify di Prampero’s approach to professional soccer players. Therefore, the aims of this study were: (1) to determine the energy cost of running on grass (C_r_) in ecological conditions on elite soccer players and (2) to validate an updated MP with a new equation using a soccer-specific exercise protocol.

## 2. Materials and Methods

### 2.1. Participants

Seventeen first-team professional male soccer players competing in the Italian Serie A (mean ± SD: age 24.0 ± 2.9 yrs, stature 175.1 ± 4.9 cm, body mass 75.9 ± 5.2 kg and $\dot{V}O_{2\max}$ 55.7 ± 3.4 mL·min^−1^·kg^−1^) were recruited for the first part of this study to determine constant-speed linear-running energy cost (C_r_). Subsequently, thirteen of these players (mean ± SD: age 22.1 ± 5.9 yrs, stature 181.8 ± 5.4 cm and body mass 76.5 ± 6.2 kg) then performed a soccer-specific intermittent exercise protocol to determine soccer energy cost (C). They were given a verbal explanation of the study protocols and provided written informed consent. All players had at least ten years of playing experience in national and international competitions. The experimental procedures were approved by the local Human Ethics Committee and complied with the Declaration of Helsinki.

### 2.2. Research Design

Prior to the experimental protocol, all players completed two sessions separated by at least 3 days. The first session consisted of an assessment of oxygen consumption ($\dot{V}O_{2}$) at rest followed by an incremental treadmill run to exhaustion to assess maximal oxygen consumption ($\dot{V}O_{2\max}$). The $\dot{V}O_{2}$ was assessed by means of a portable breath-by-breath gas analyzer (Cosmed K4b^2^, Rome, Italy). During the second session, each player performed a run on a UEFA standard grass soccer pitch with soccer shoes to evaluate C_r_. After this, thirteen players performed soccer-specific intermittent exercise protocol to assess C. Participants body mass and stature were assessed to the nearest 0.1 kg using weighing scales while the stature, to the nearest 0.01 m, using a stadiometer (weighing/stadiometer: seca GmbH & Co. KG., model 217, Hamburg, Germany) during the first visit.

### 2.3. Maximal Oxygen Consumption ($\dot{V}O_{2max}$) Assessment

The test started at 8 km·h^−1^ and increased by 2 km·h^−1^ every 2 min until 16 km·h^−1^ was reached, after which the gradient was elevated by 2.5% every 2 min until volitional exhaustion. Continuous $\dot{V}O_{2}$ and H in each 5s period was measured. $\dot{V}O_{2\max}$ and maximal heart rate (HR_max_) were identified as the peak value at volitional exhaustion [28].

### 2.4. Constant-Speed Linear-Running Assessment

The C_r_ assessment was performed on a UEFA standard grass soccer pitch (Artemio Franchi stadium, ACF Fiorentina, Florence, Italy) and required all players to wear soccer shoes. Participants were asked to perform a 6 min aerobic-based steady-state, run at 10.29 km·h^−1^ (73% $\dot{V}O_{2\max}$) around a 160 m circular course with cones set at 20 m interval running speed was regulated sound signal emitted every 7s to maintain the required 10.29 km·h^−1^ (Figure 1). Participants’ heart rate (b·min^−1^) was determined by a chest-based telemetry monitor (RS800sd, Polar Electro Oy, Kempele, Finland). The $\dot{V}O_{2}$ was measured on a breath-by-breath basis using a portable analyzer (K4b^2^, Cosmed, Rome, Italy) calibrated in accordance with the manufacturer’s instructions. During the constant-speed linear-running test, the portable unit was worn in a harness on the players’ shoulders. Data were recorded locally by the portable unit and sent telemetrically to a PC. The $\dot{V}O_{2}$ data were averaged over 1 min. Steady-state $\dot{V}O_{2}$ was defined as the mean of the last 3 min of the constant-speed linear-running test. All participants were fully familiarised with the instruments and procedures.

C_r_ was calculated as the ratio between $\dot{V}O_{2}$ above its resting value ($\dot{V}O_{2n}$) and the speed (*v*):$$C_{r}=\dot{V}O_{2n}/v$$
assuming an energy equivalent of 20.9 J per mL of O_2_ (corresponding to a non-protein respiratory exchange ratio of 0.96) and *v* is 10.29 km·h^−1^ (i.e., 2.86 m·s^−1^), a speed below the anaerobic threshold one. When a steady-state exercise is performed below the anaerobic threshold intensity, it is assumed the metabolic energy demand is satisfied by means of the aerobic energy mechanism. The C_r_ calculated is, therefore, expressed in J·kg^−1^·m^−1^.

### 2.5. Soccer-Specific Running

Prior to data collection, all players performed a $\dot{V}O_{2\max}$ identical to the test used during the first aim of the study (see above). Players then underwent several familiarising runs with the varying running speeds used in the soccer-specific intermittent exercise protocol through sound dictation emitted prior to physiological data collection. All players were required to perform a soccer-specific run for 8 min at varying speeds. During each trial, participants were fitted with a 10 Hz GPS unit (BT-Q1000eX 10 Hz, Qstarz, Taipei, Taiwan), a heart rate monitor and a portable gas analyzer (K4b^2^ Cosmed, Rome, Italy) to assess distance and speed, heart rate, and oxygen uptake, respectively. At the start and end of each testing session, blood lactate (La^−^)b was collected via a portable lactate analyzer (Lactate Pro^TM^ LT-1710, Arkray Inc., Kyoto, Japan). The simulation protocol to determine C of soccer-specific movements on a UEFA standard grass surface was undertaken with soccer boots at the Artemio Franchi stadium (ACF Fiorentina, Florence, Italy).

### 2.6. New Soccer-Specific Energy Cost Equation

The new C equation modified, according to Minetti [17], was derived from findings in the first part of the study using Pearson’s correlation coefficient modeling of C_r_. A C_r_ value of 4.66 J·kg^−1^·min^−1^ was established using several curve models from a 2nd to a 4th order polynomial fit. The determination of this new C_r_ as the constant term is central to the identification of MP in soccer and further enhances previous models by providing both a surface-specific and population-specific energy constant. The previous constant of 3.6 J·kg^−1^·min^−1^ was altered to reflect the measured C_r_. The incorporation of a new constant disrupts the essential mathematical slope and shape of the Minetti et al. [17] model ranging from −45 to 45%, and impairs the calculation of MP. To maintain the integrity of the equation in this range (−45 to 45%) while correcting for the underestimation of C that is determined during decelerations (gradients) greater than −45%, modification of the slope (% gradient)/energy cost data with a different model were used. Following the mathematical model of di Prampero et al. [16], fitting acquired data with a 4th order polynomial (Figure 2) meets the requirements of the equation. The constant term of the C_r_ is held at 4.66 J·kg^−1^·m^−1^, and the error term is minimised (0.12 J·kg^−1^·m^−1^) more efficiently than either a cubic or parabolic model. Decreasing the order of the polynomial fit for the equation to both a 2nd and 3rd-order (2°–3° degree) translates the trend line, shifting it upwards on the *y*-axis, in effect overestimating the directly determined constant term from 4.66 to 4.79 J·kg^−1^·m^−1^. The new “C equation” was raised to the fourth-degree polynomial. In doing so, it does the following: (a) remove the negative inflexion point associated with deceleration activity over a −45% gradient apparent in the original Minetti et al. [17] model; (b) the C_r_ of the constant speed running remains comparable to that directly measured in the 17 elite soccer players. The C_r_ (known term) of 4.66 J·kg^−1^·m^−1^ was used for the subsequent analysis below.

### 2.7. Determination of C on Grass: Soccer-Specific Intermittent Exercise Protocol

All players were required to perform a soccer-specific intermittent exercise protocol consisting of 8 min at varying speeds (Table 1). The protocol incorporated various actions, from maximal to sub-maximal sprints, shuttles with changes of direction (CoD) of varying angles. It included slaloms around cones, including short passive and long passive recoveries from the maximal sprints. The protocol was divided into 4 phases and repeated for 8 laps to give a total activity duration of approximately 8 min (Figure 3). All activities commenced from a stationary start position and required participants to come to a complete stop at the end of the marked phase. Upon completion of each lap, the next lap started at the end position of the previous one (the opposite side), always mirroring the same sequence of activities (Figure 3). A trundle wheel was used to measure the exact length of the circuit. In addition, the angle of movement was measured and set at ±30° during setup. During the circuit, an iPod system (iPod nano, Apple, Cupertino, California, USA) was used to emit the pacing beep every 5s. As a spatial reference, multiple markers were positioned at fixed points depending on the running speed required. Each subject was given a heart rate monitor (RS800sd, Polar Electro Oy, Kempele, Finland). The $\dot{V}O_{2}$ was determined on a breath-by-breath basis using a portable analyzer (K4b^2^, Cosmed, Rome, Italy). The metabolic unit was calibrated using a 3-L syringe and a gas of known composition (16.00% O_2_, 5.00% CO_2_). During the steady-state run, the K4b^2^ analyzer was placed in the harness around the shoulders of the participants. Before the start of the study, all subjects were familiarised with the equipment and the procedures. The data were recorded by the central unit located in the harness and sent telemetrically to a PC. The $\dot{V}O_{2}$ data were averaged over 1 min. The [La^−^]b concentration was determined using a portable analyzer (Lactate Pro^TM^ LT-1710; Arkray Inc., Kyoto, Japan) using a blood sample obtained from fingertip upon completion of the test. Steady-state O_2_ uptake was defined as the mean of the last 3 min of the constant speed run.

### 2.8. Calculation of Energy Cost (C) and Metabolic Power through Direct Physiological Measurement ($P_{\dot{V}O_{2}}$)

The C was determined 24 h after the last training bout and ~2 to 4 h after the last meal; it was evaluated during soccer-specific protocol performance. The C on the soccer-specific activity protocol was calculated from the ratio of the total metabolic energy expenditure (EE in joules) utilised above resting to the distance covered (*d* in m). The C above resting was calculated from the sum of the Aerobic (Aer), Anaerobic Alactic (AnAl), and Anaerobic Lactic (AnL) energy expenditure. Briefly, EE derived from aerobic sources was obtained from the integral from the onset of exercise to the end of the soccer-specific protocol; the net $\dot{V}O_{2}$ values (averaged over 60s), as obtained directly during the test minus the pre-exercise resting $\dot{V}O_{2}$ values (~3.5 mL·kg^−1^·min^−1^). Furthermore, the contribution from anaerobic AnAl energy expenditure was determined by the assessment of the $\dot{V}O_{2}$ uptake determined during the first 6 min of recovery upon completion of the protocol. The net $\dot{V}O_{2}$ values obtained from the 4th to 6th min of recovery were used to estimate the fast alactic O_2_ debt. Finally, the lactic contribution to the overall energy expenditure (AnL) was estimated after exercise from the net [La^−^]b accumulation above resting, by an energy equivalent of [La^−^]b accumulation in blood equating to ~3 mL·kg O_2_ per mM [29,30]. The overall EE (= Aer + AnAl + AnL) for the duration of the test (8 min) was determined. This value of $\dot{V}O_{2}$ was multiplied by 20.9 based on the assumption that 1 mL O_2_ yields 20.9 J, divided by the mass of the subject (kg) and distance covered (m) to generate an estimate of soccer-specific protocol C in J·kg^−1^·m^−1^. Data were converted to W·kg^−1^ using the following equation: MP (W·kg^−1^) = C·*v*
where C is energy cost and *v* running velocity. Directly determined estimates were derived and compared with the C (and W·kg^−1^) calculated from the new energy cost equation.

### 2.9. Indirect Calculation of Metabolic Power through GPS (P_GPSn_ and P_GPSo_)

Participants were tracked over the linear-sprints and the soccer-specific circuit using a GPS device (BT-Q1000eX 10 Hz, Qstarz, Taipei, Taiwan). Instantaneous velocity measurements were obtained for each trial. The GPS unit was placed on the upper back between the shoulder blades in a custom-made vest on all players. According to Witte and Wilson [31], the mean ± SD number of satellites “used” during data collection was 9.0 ± 0.5 (satellites “tracked”: 11.0 ± 0.4). The mean horizontal dilution of precision (HDOP) during data collection was 1.0 ± 0.1. GPS velocity data (10 Hz) was sampled and synchronised at the first movement recorded above 0 m·s^−1^ to account for processing phase delays within the breath-by-breath output on the Cosmed K4b^2^. Data were downloaded and analysed (GPS Metabolic Power LagalaColli v9.076d, SPINItalia, Roma, Italy) to establish the time, speed, and distance. The MP was determined according to each player’s individual body mass through the methodology of C modeling as previously described and modified according to the new energy cost equation [16,17].

### 2.10. Statistical Analysis

All data were analysed using Statistical Package for the Social Sciences version 25 for Windows (SPSS, Chicago, IL, USA). Results are expressed as mean ± SD after verifying the assumption of normality of the distributions of the dependent variables using the Shapiro–Wilk test. Differences between methods were compared in accordance with the recommendations of Ludbrook [32] for the calibration of one method against another. Pearson’s correlation coefficient was assessed to establish whether linear relationships between both methods were present. In accordance with Ludbrook’s [32] recommendations, to gain estimates of fixed (intercept) and proportional (slope) bias, the coefficients of the ordinary least products regression were determined via the LOSS function in SPSS version 25 (see Ludbrook [33,34] for SPSS LOSS function commands). The SPSS LOSS function also provided the 95% CIs (via bootstrapping) for the intercept and slope along with the predicted and residual values for the raw data. The 95% prediction intervals were then calculated in the manner described by Altman et al. [35]. The estimates of MP are represented by the median ± interquartile range (IQR). The alpha level of significance was set at 5% [36]. To assess MP (average for all bouts) differences over the 3 different conditions ($P_{\dot{V}O_{2}}$ − P_GPSn_ − P_GPSo_), repeated-measures analysis of variance (ANOVA) was used. When a significant *F*-value was found, post hoc analysis between conditions was performed using Fisher’s least significant difference correction procedure (Bonferroni).

## 3. Results

### 3.1. Energy Cost of Constant-Speed Linear-Running (C_r_) and $\dot{V}O_{2max}$

The mean ± SD for $\dot{V}O_{2\max}$ values in 17 elite soccer players were 55.7 ± 3.4 mL·kg^−1^·min^−1^. The new C_r_ on a UEFA standard grass soccer pitch in elite players was calculated and determined as 4.66 J·kg^−1^·m^−1^ (Table 2).

### 3.2. Energy Cost of Soccer-Specific Running (C)

#### Method Comparison P_GPS_ and $P_{\dot{V}O_{2}}$

Table 3 shows the estimates of fixed (intercept) and proportional (slope) bias for both P_GPSn_ and P_GPSo_ relative to $P_{\dot{V}O_{2}}$ [17]. Pearson’s correlation coefficient between P_GPSn_ and $P_{\dot{V}O_{2}}$ (Cosmed K4b^2^) was 0.66 (95% CI = 0.19 to 0.88) and between P_GPSo_ and $P_{\dot{V}O_{2}}$ 0.63 (95% CI = 0.16 to 0.87). Along with the scatterplots, the *r* values were considered as enough evidence for the presence of a linear association for both methods with the $P_{\dot{V}O_{2}}$ (gold standard). The scatterplots for P_GPSn_ on $P_{\dot{V}O_{2}}$ and P_GPSo_ on $P_{\dot{V}O_{2}}$ showed no evidence of heteroscedasticity (Figure 4). The new P_GPS_ model indicates that all variables are distributed around the line of unity, indicating that it will both under and overestimate MP relative to the P_GPS_ of Minetti et al. [17], which routinely underestimates MP across all ranges.

### 3.3. Physiological Response, Locomotor and Metabolic Demands of the Soccer-Specific Circuit

The mean ± SD for $\dot{V}O_{2\max}$ and HR_max_ values attained during the laboratory treadmill tests were 61.1 ± 4.3 mL·kg^−1^·min^−1^ and 194.0 ± 6.0 b·min^−1^, respectively. Mean ± SD values of $\dot{V}O_{2}$ and HR_max_ elicited during the soccer-specific circuit performance were 44.5 ± 2.3 mL·kg^−1^·min^−1^ (~73% $\dot{V}O_{2\max}$), and 182.0 ± 8.0 b·min^−1^ (~94% of HR_max_), respectively. Mean ± SD [La^−^]b values were 7.2 ± 1.6 mmol·L^−1^ post-soccer-specific protocol performance, with a net increase of 6.2 ± 1.6 mmol·L^−1^ compared to resting values (Table 4).

For illustrative purposes, the soccer-specific circuit data, relative to 60s or one complete lap, with the exclusion of the other three laps, is displayed to highlight the onset in $\dot{V}O_{2}$ present and show the four locomotor activity phases (Table 1). To obtain the $\dot{V}O_{2}$ data related to body weight, the mL·min^−1^ for the kg of each subject were divided accordingly (Figure 5).

The results of the locomotor response of the players during the soccer-specific circuit performance can be found in Table 5. The other seven parameters evaluated in relation to the external load are displayed in Table 6.

### 3.4. Metabolic Power: P_GPSn_ and P_GPSo_ vs. $P_{\dot{V}O_{2}}$

The MP (W·kg^−1^) of the soccer-specific circuit was indirectly determined from GPS locomotor data and estimated through two energy cost equations: the original Minetti et al. [17; P_GPSo_] and the present modified equation derived from Minetti et al. [17; P_GPSn_]. Each equation was “calibrated against” direct gas exchange and blood lactate data during the soccer-specific protocol performance using ordinary least product regression. The correlation coefficient between P_GPSn_ and $P_{\dot{V}O_{2}}$ was 0.66 (95% CI = 0.19 to 0.88) and between P_GPSo_ and $P_{\dot{V}O_{2}}$ was 0.63 (95% CI = 0.16 to 0.87). Along with the scatterplots, the *r* values were considered as evidence for the presence of a linear association for both P_GPSn_ and P_GPSo_ relative to $P_{\dot{V}O_{2}}$ (calibrator method). The scatterplots for P_GPSn_ on $P_{\dot{V}O_{2}}$ and P_GPSo_ on $P_{\dot{V}O_{2}}$ showed no evidence of heteroscedasticity (Figure 4). The intercept in both regression equations provides an estimate of the amount of fixed bias between the values derived from P_GPSn_ and P_GPSo_ in relation to $P_{\dot{V}O_{2}}$, and the slope of the regression line determines the amount of proportional bias. In both cases, the estimates of fixed bias are negligible (P_GPSn_ = −0.80 W·kg^−1^ and P_GPSo_ = −1.59 W·kg^−1^), and the bounds of the 95% CIs show that they are not statistically significant from 0. The estimates of proportion bias are also negligible as they are both close to one (the absolute differences from one being 0.03 W·kg^−1^ for P_GPSn_ and 0.008 W·kg^−1^ for P_GPSo_), and they too are not statistically significant as both 95% CIs span 1. Although the intercept and slope of the ordinary least products regression can be used to estimate fixed and proportional bias, the impact of the scatter in the values also needs to be determined when assessing the suitability of the GPS equations. To this end, the 95% prediction intervals for P_GPSn_ and P_GPSo_ are included in Figure 6 as these provide a plausible range in which a player’s P_GPSn_ or P_GPSo_ value may actually lie within for a given $P_{\dot{V}O_{2}}$ value. For example, if a new player is taken from the same population and has their $P_{\dot{V}O_{2}}$ measured as 15.5 W·kg^−1^ through the direct measurement approach, then by using the intercept (−0.803) and slope (1.030) estimates from the new energy cost**** equation,**** their predicted P_GPSn_ value would be 15.16 W·kg^−1^ with a 95% prediction interval of ± 1.52 W·kg^−1^. This means it is plausible that their actual P_GPSn_ value when predicted using the new energy cost equation**** could be as low as 13.64 W·kg^−1^ or as high as 16.68 W·kg^−1^. Similarly, using the Minetti et al. [17]**** equation for the same player, their predicted value would be 13.78 W·kg^−1^ and 95% prediction interval would range from 12.28 W·kg^−1^ to 15.30 W·kg^−1^. Repeated-measures ANOVA showed differences over MP conditions (*F*_1,38_ = 16.929 and *p* < 0.001). Following Bonferroni post hoc test showed significant differences regarding the MP (Figure 6) between P_GPSo_ and $P_{\dot{V}O_{2}}$/P_GPSn_ (*p* < 0.001), while no differences were found between $P_{\dot{V}O_{2}}$ and P_GPSn_ (*p* = 0.853).

Figure 7 compares $P_{\dot{V}O_{2}}$ and P_GPSn_ during the 8-lap of the soccer-specific circuit. Unlike the study performed by Buchheit et al. [24], it was not possible to separate MP during the recovery phases within the soccer-specific circuit as these occurred intermittently within each minute (lap) and not at the end of each lap. In addition, the recovery period time was shorter and more times compared to recoveries used in other studies (Table 1). Only the bars relating to the integral minutes of work are shown, including the short passive recovery (Figure 7). $P_{\dot{V}O_{2}}$ respects the physiology of physical exercise [34] by adding in the calculation of the estimated energy expenditure (EEE), the energy obtained from the O_2_ debt (AnAl and AnL systems).

## 4. Discussion

The first aim of the present study was to determine the C_r_ of elite professional soccer players in ecological conditions, which was found to be 4.66 J·kg^−1^·m^−1^. Previous findings have observed that O_2_ consumption measured during constant-speed linear-running amounts to 73% of maximum O_2_ consumption, thus confirming the aerobic nature of the metabolic demand due to the set C_r_ assessment speed used in this study. Pinnington and Dawson [18] and Rodio et al. [37] have reported values of 4.64 and 5.70 J·kg^−1^·m^−1^ when running on natural grass in a group of recreational runners and sedentary males, respectively. In amateur soccer players, Sassi et al. [38] reported a C_r_ value of 4.20 J·kg^−1^·m^−1^, which is slightly lower than the value determined in the current study. More recently, Stevens et al. [22] determined C_r_ be approximately 4.6 J·kg^−1^·m^−1^ at a running speed of 10 km·h^−1^ on artificial turf in a group of non-elite soccer players. Present data differs from previous findings with differences in C_r_ estimates potentially due to variations in the population (elite vs. non-elite), playing surface (artificial vs. non-UEFA grass) and use of footwear, which affect running kinematics and energy cost [39]. It is also important to note that the C_r_ presented herein reflects elite players in a non-fatigued state. It remains to be determined whether the C_r_ changes as a function of fatigue-related changes in metabolic, biomechanical and neuromuscular efficiency fluctuations during match-play and training.

Having established a new C_r_ in elite professional soccer players of 4.66 J·kg^−1^·m^−1^, a further aim was to incorporate this finding into and validate a new MP algorithm that includes a specific C term. This was established on elite soccer players, on an elite UEFA grass playing surface, with soccer-appropriate footwear to best replicate factors deemed important in the determination of movement economy in this population. This was facilitated through the use of the new constant term of C_r_ and regression prediction equation for the assessment of the C, by using the average MP on a soccer-specific test through direct and indirect measurements of O_2_ consumption using di Prampero’s et al. [16] approach while modifying Minetti’s et al. [17] equation of energy cost. The validation takes into account locomotor kinematic data for the P_GPSn_ calculation, which is as soccer-specific as possible. The method has developed progressively through the work of di Prampero et al. [16] and Osgnach et al. [13], together with the calculation of the C [17]. These approaches cannot be applied to other sports, but only for soccer which has a “predominately horizontal” mode, where running on flat terrain represents the largest portion of the energetic performance model.

Stevens et al. [22] described that it is currently not clear to what extent it is possible to compare MP and C between varying running activities. Therefore, the validation of the different running activities is the main goal of this research. There are some criticisms and limitations present within the current research studies available in the literature because of previous protocols proposed and utilised. Present findings established that shuttle running using low speeds only, thus negating maximal actions, negatively affects the P_GPS_. Current P_GPS_ observations are lower than the values found in other studies [22,25] due to the decreased error using 10 Hz sampling frequencies, thus preventing a drastic decrease. Further, that type of running is not ideal or specific for amateur soccer players. The proposed speeds have an influence on cost when establishing values of P_GPS_ (due to the absence of sub-maximal bouts) and are very much in direct relation with $P_{\dot{V}O_{2}}$, therefore amplifying the difference between the indirect measure and gold standard [22].

Highton et al. [26] observed that during a rugby-specific protocol incorporating contacts (collisions) and extensive passive rest (standing recovery), these measures were directly responsible for the underestimation of the EEE from the GPS data. The lack of agreement between direct and estimated values was deemed to be a result of the inability of GPS to detect EE associated with non-locomotor exertion. Brown et al. [27] previously examined the validity of a GPS tracking system to estimate EE during exercise and field-sport locomotor movements in healthy adults, but differences in population sample make comparisons related to C observations with our study difficult. In addition, a GPS system that interpolates (via accelerometers) 5 Hz data is deemed insufficient for the speed variations present in the “modern” soccer game and is therefore not comparable to our results [10]. Finally, the incorporation of extensive recoveries in their exercises, which are then used for the calculation of EEE, result in an increase in the differences between $P_{\dot{V}O_{2}}$ and P_GPS_, underestimating the latter. In addition, the failure to collect [La^−^]b measurements is a further limitation.

Buchheit et al. [25] used a test incorporating a ball within their assessment protocol, which increases the specificity related to soccer. However, they did so without considering the proportion of time a player is in possession of the ball during a game, making this a key factor causing P_GPS_ underestimation. Findings show that soccer players are in possession of the ball for less than 1% of the total playing time and less than 2% of the total distance they travel during a match [40,41]. The C of running with the ball is higher than without the ball, meaning that the $\dot{V}O_{2}$ will rise during the technical parts and other activities with the ball [25,42]. The P_GPS_ will always be underestimated when incorporating an assessment protocol focusing on technical aspects with the ball since the kinematic data assessed is closely related to the exercise performed. Therefore, a lack of high-speed, acceleration and deceleration movement patterns adversely affects P_GPS_ estimation. Potentially, the decelerations will be further emphasised (e.g., stop and kick, ball control during a slalom, passing and reception with a rebound wall, etc.) which from a metabolic point of view (P_GPS_) has a low C. Additional limits are present such as the use of a 4 Hz GPS sampling frequency, the intensity percentage of the $\dot{V}O_{2\max}$ which was found to be up to 64% despite the “low” mechanical demands and the absence of high-speed (>14.4 km·h^−1^) movement patterns as previously observed in detail by Osgnach et al. [43].

Considering all of the research which has been conducted, it is important to consider the previous limitations observed in order to create/develop a test that can validate P_GPS_ in soccer and be compared to $P_{\dot{V}O_{2}}$. It has been found that three fundamental macro-aspects are required to ensure that the energy estimation method proposed by di Prampero et al. [16] and Osgnach et al. [13] best represents the soccer game and these are classified as follows:(i)using a GPS system with a minimum sampling frequency of 10 Hz and a mathematical reduction (or smoothing, e.g., moving mean) of the speed data at 5 Hz to reduce noise in GPS elevation data. This has been found to be a methodologically verifiable value as established in a study performed by Gaudino et al. [44], through performing calculations on contact and flight times in players on natural grass;(ii)using an appropriate method to calculate the MP in an intermittent sub-maximal exercise, including the anaerobic amount [30,45];(iii)using an appropriate experimental design by ensuring the work protocol includes an adequate population (elite soccer players), a specific terrain (natural grass) and appropriate footwear (soccer shoes) for calculating a specific C. Further, the performance model, such as the work: rest ratio and different locomotor activities (e.g., sprinting, walking, jogging, high-speed running, CoD, etc.), must be suitable to the activity as the duration of pauses/recoveries plays a decisive role on the metabolic response in soccer.

With regard to creating a test, other precautions related to the relationship between total high speed (TS) and total high-power (TP), which has been found to be between 55% and 32% in favor of TP [46,47], and the dimension changes as we move from SSGs’ (small-sided games) scenarios to the whole field (105 × 68 m), must also be considered. Therefore, choosing a soccer-specific circuit with an average MP greater than the 11–12 W·kg^−1^ game intensity [48] is necessary to respect the methodological criterion of the power–time relationship on intermittent exercises [49]. Furthermore, the high-intensity actions (>20 W·kg^−1^) are found to be around 4.7 actions per min (Table 6), a value that is higher than previous findings where only 2 intense actions per min have been described in studies that only considered speed thresholds [50,51]. In addition, O’Donoghue [50] shows that the most frequent recoveries are <30 s, and about 57% of those are <20 s. It is important to note that resting periods are never cumulative (pause of 30s consecutive) but fractioned in the game and must therefore also be incorporated in this way. Further, Bradley et al. [52] showed that recovery time, defined as the time that elapsed between high-intensity running actions, is about 52 ± 18 s. For this reason, in our soccer-specific intermittent exercise protocol, a maximal action (triangle) is repeated every single lap (1 min). If you compare this to work conducted by Buchheit et al. [25], where the passive recovery of 30 s was entirely spent at the end of the work minute, our proposed model of the circuit respects the intermittent nature of soccer by dividing the ~25 s of total recovery into shorter breaks (<10 s).

According to Buchheit and Simpson [23], Malone et al. [53] and Varley et al. [54], acceleration is measured from GPS data mostly derived from the doppler-shift velocity. The time interval over which acceleration is calculated can significantly alter the data with a wider interval resulting in a smoothing effect on the data. Typically, acceleration is calculated over 0.2 or 0.3 s when using 10 Hz GPS, although the most appropriate interval will depend on the brand and the model of the device [53,55]. In the present soccer-specific intermittent exercise protocol, it was preferred to export raw data from commercial software and process it independently. The method used to calculate the high accelerations, using the kinematic data of official Serie A matches (Savoia et al. unpublished data) as a reference database, is based on the equation by Sonderegger et al. [56] and modified accordingly:a_max_ (max acceleration): −0.18 (−0.17 to −0.19)·*v_init_* + 5.91 (5.80 to 6.02)
where a_max_ is expressed in m·s^−2^ and the *v_init_* in km·h^−1^ (Table 6).

The conceptual basis of the new MP approach to soccer, as initially described by Osgnach et al. [13], was predicated upon a formula that was based on a population non-specific to soccer. The data reported in this study was assessed through C_r_ of running on grass using a new version of the “C equation” (modified equation [17]). It was then applied to a soccer-specific high-intensity protocol for the first time relative to that observed in the original equation. The principal finding of this study directly determined the physiological demands simultaneously with GPS derived modeling of the MP. This indicates that in an elite soccer population, the GPS metabolic power paradigm provides similar point estimates of determining work rate during the soccer simulation protocol/activities undertaken. Specifically, data shows that where there are periods of repeated accelerations and decelerations, CoD when superimposed upon an aerobic background MP estimates were approximately similar between the direct ($P_{\dot{V}O_{2}}$) and indirect approaches (P_GPSn_ and P_GPSo_; Figure 6). Comparisons between both equations demonstrate that both effectively estimate MP with the P_GPSn_ having a marginal advantage over the P_GPSo_. The former has a marginally lower fixed bias and similar proportional bias suggesting that as MP increases, the error remains relatively constant, i.e., no heteroscedasticity suggesting it works across the range of running activities and speeds incorporated into the protocol. The widths of the prediction intervals are also similar. In order to improve the prediction interval precision, a larger validation in a similar elite population would be needed. Interestingly, MP during the first phase of the soccer-specific protocol was higher, possibly due to the higher metabolic demand associated with the acceleration phase of running. It has previously been determined that a very low metabolic demand is associated with phases of deceleration during nonlinear runs and that acceleration or re-acceleration phases display an increased metabolic requirement [57].

Further, we show that using a high-frequency GPS system sampling at 10 Hz, movement patterns are subject to rapid CoDs and that the C paradigm still provides a representative metabolic formula that is optimised to elite soccer match-play and training. The accuracy of the GPS estimates is based upon higher sampling frequency and accuracy for both acceleration and deceleration phases of soccer-specific movement. Present data are also indicative of the essentially aerobic nature of soccer-specific movement patterns. The protocol as applied in this study elicited a physiological strain approximating just over 70% of $\dot{V}O_{2\max}$ and 90% of HR_max,_ which is in line with that reported during match-play [41,58]. Such observations provide support for the soccer-specific model utilised in the present study to compare directly and indirectly measured MP and reflect its specificity in relation to soccer match-play.

Applying a model that evaluates matches and incorporates training variables through biomechanical and energy intuition is easy to apply when the kinematic data are available. Through the study of $P_{\dot{V}O_{2}}$ and P_GPS_, it has been possible to widen the vision of the external training load by rationally integrating the information derived from speed alone with those related to its variations over time. $\dot{V}O_{2\max}$ explains only 24% of P_GPS_ (*r* = 0.49), thus suggesting that players with a higher maximal O_2_ consumption consequently have a higher MP. However, such an assumption would lead to an assessment error, given that there are subjects capable of expressing a P_GPS_ of ~15.7 W·kg^−1^ with a maximal O_2_ consumption of 70 mL·kg^−1^·min^−1^ and 57 mL·kg^−1^·min^−1^, which equates to a 23% reduction/difference in aerobic power. Therefore, a high $P_{\dot{V}O_{2}}$ does not necessarily imply a high P_GPS_, which represents the external load: the effect on the speed data calculated through the movements tracked by the GPS.

Ultimately there are three issues that need to be methodologically expanded:(i)the extra energy, reasoning on the use of 100 Hz tri-axial accelerometers together with the correct mathematical filters. Buchheit and Simpson [23] addressed this discourse by arguing that accelerometers are practical assessment methods to quantify stride variables when used indoors (i.e., no GPS signal is required), therefore allowing the use of these for intermittent team-sports (e.g., basketball, handball, etc.). All of this could improve the assessment of eventual muscle strength deficits in players, leading to progress in the field of injury recovery [23,59]. Further, Osgnach et al. (unpublished data) are currently focusing on a study related to “Muscle Power” (GPEXE ©, Exelio Srl, Udine, Italy), with the aim of considering the greater muscular load of the braking activities (decelerations) compared to those observed during accelerations. Findings could reduce the underestimation of P_GPS_ compared to $P_{\dot{V}O_{2}}$ by considering and evaluating the addition of a small energy surplus deriving directly from neuromuscular fatigue. Technologies such as surface electromyography in correlation with current estimates of MP could be decisive for developing new C equations (with more attention to, e.g., decelerations, CoD, etc.) according to Buchheit et al. [59] and Hader et al. [57]. The main reason why it was chosen not to implement the update to the concept of equivalent slope [60], in addition to the changes proposed by di Prampero and Osgnach [61] on the inclusion of a lower energy cost for the walking phases (C_w_), is dictated by the fact that the original equation [17], with small adjustments as presented in Figure 2, is already able to estimate the EE of an intermittent exercise albeit within a confidence interval range [62].(ii)The performance model in the choice of tests that we want to validate together with the calculations on the energetics of muscular exercise. In support of this, Brown et al. [27] mentioned that using other criterion procedures that can measure both anaerobic and aerobic EE directly can help with the assessment of validating the approach, a concept found to positively work in this study. Further incorporations could have been made to help improve the current study: (1) the insertion of the ball in the soccer-specific circuit for a maximum of 5 to 10s per lap (e.g., sprinting with the ball or 5s of ball control and passing or shooting, etc.); (2) the inclusion of some walking/slow running phases given the active nature of the recovery (5–10 W·kg^−1^) in soccer; (3) an increase of the sample studied in order to statistically understand the relation of the P_GPS_ when compared to the $P_{\dot{V}O_{2}}$ measurement; (4) the addition of a camera at the start to record the time during the maximum triangle performed at each lap, in order to obtain a series of times to assess the min-by-min performance decrement [63,64]. This “new” test could be an alternative to the various repeated sprint ability tests previously used in the literature [65,66,67], as it alternates the maximal bouts with runs and recoveries of various kinds, thus better simulating the intermittent scenario of the game. These concepts are closely linked to the decisive role of the C, which was shown to be ~38% higher than C_r_ on the grass at a constant speed (6.41 J·kg^−1^·m^−1^ > 4.66 J·kg^−1^·m^−1^, Table 2 and Table 4) in our soccer-specific intermittent exercise protocol. Therefore, it is essential that the training plan is soccer-specific and focuses on the economy of movements throughout the gameplay (training drill), in order to not obtain a greater efficiency of the running technique (i.e., athletics), as this would lead to an improvement of the C_r_. Observations by Buglione and di Prampero [21] found C_r_ to get worse during a competitive soccer season, highlighting the need for tests and training to be sport-specific, respecting the biomechanics related to running in soccer.(iii)The application of the “MP” approach to video match- and time-motion analysis using the same equations and algorithms as used by the GPS software for training load analysis represents the future of soccer. This would enable coaches and practitioners the possibility of assessing the metabolic performance of each player during every match and help with the study of trends related to the loads incurred during training and gameplay. The uniformity and homologation of the algorithms would really represent a “turning point” to compare training methodologies/philosophies (i.e., traditional, integrated, structured, tactical periodization, etc.) and similarities and/or differences between championships and competitions (with respect to the presence or absence of cup tournaments). Its usefulness could further extend on the choice of purchasing the appropriate player(s) during the transfer window by helping to update databases which are useful for soccer scouting and match analysis (such as, Wyscout, InStat, Stats Perform, Transfermarkt, etc.). Further, combining the technical-tactical information, the physical performance and the history of injuries provides a better understanding of the players’ official performance parameters and the current training loads carried out at their club through the use of “integrated soccer language” [68].

However, despite some of the current limitations, as well as the sample, the information provided can help better describe the activity currently present in the game of soccer and its concept related to intensity by providing a “new” opening to the scenario of analysis and observation. Player practice on the field imposes the application of these models to the daily activity of analysis of training sessions, both with and without the ball. In the last few years, most games have almost total coverage of video tracking systems in the stadiums and have the possibility of applying high-frequency GPS to the players during their weekly microcycles. The comparison and the periodization of training have become a great interest among the professionals working within this sector. A modern-day data scientist who wants to get closer and closer to fully understanding the game cannot merely untie the purely physical data (response) from the technical-tactical aspect (cause) during match-play. Considering soccer is a situational sport, its performance is mainly influenced by the need for strategical and tactical developments in the field, which could also impose moments of pause/rest between repeated high-intensity actions.

In the future, it would be desirable to use larger data sets through machine learning and data-mining systems so that the spectrum of player-specific analysis can be further expanded. This will help with the association of not only the “static information” to the physical data, such as player position, the system of play, result, percentage of ball possession, etc. but also the “dynamic information”, such as the flow of tactical attitudes tagged through video match-analysis. Further research will provide information that helps with the understanding and the explanation of the observed physical data in accordance with specific moments/actions during a game. Ultimately, it could help managers/coaches choose a “winning” strategy.

## 5. Conclusions

The results of this study report the energy cost of running in a soccer-specific manner. Respecting the performance model is the cornerstone for developing a test, which takes into account the intermittent nature of soccer. In order to assess the efficiency of movements that occur in soccer, the test must incorporate all the variables, which are able to influence the energy cost (population, terrain, footwear, type of accelerations/decelerations, etc.). Taking these variables into account, an “updated” energy cost equation in elite soccer players that derives an estimate of metabolic power from kinematics data closer to the “gold standard” approach is presented. In using the statistical approach, it can be suggested that the P_GPSn_ could substitute P_GPSo._ These findings are useful for practitioners and coaches in understanding the error associated with using the metabolic power approach in training and match-play training load estimates and the error associated with that in elite players.

## Figures and Tables

**Figure 1 ijerph-17-09554-f001:**
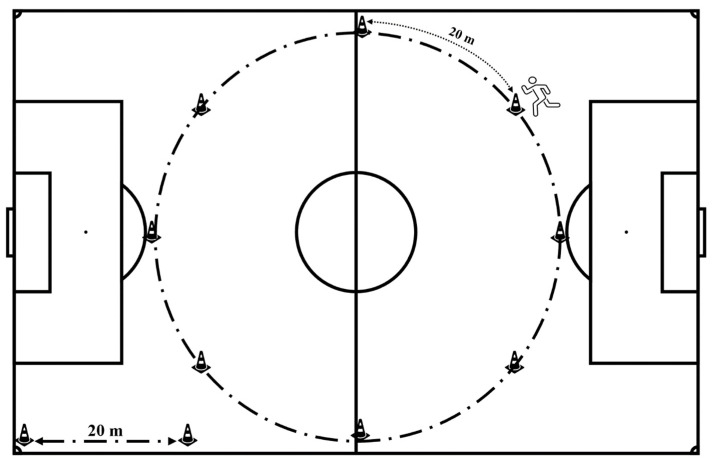
UEFA standard grass soccer pitch adapted for a constant-speed linear-running test.

**Figure 2 ijerph-17-09554-f002:**
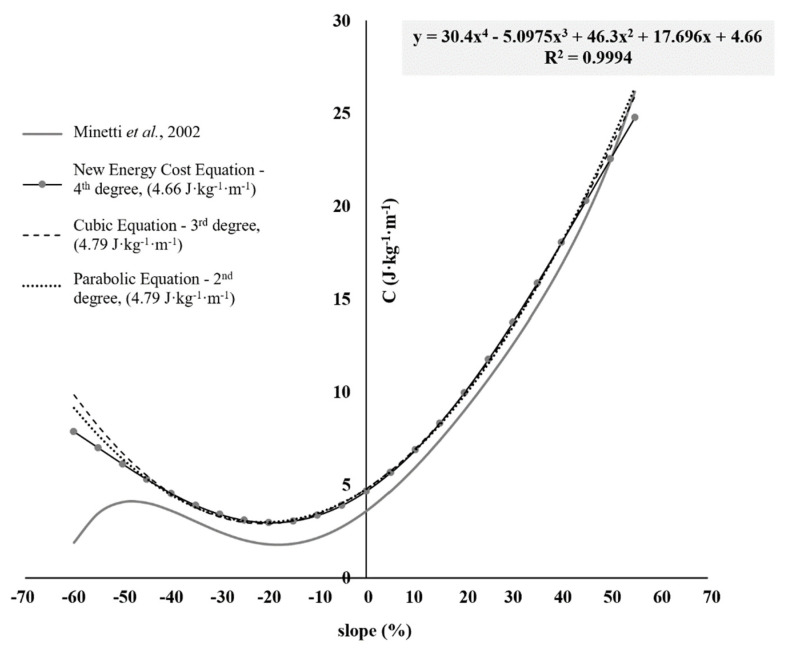
New energy cost paradigm (4th order polynomial fit) relating to the C of running over grass in elite soccer as a function of the gradient with initial C_r_ constant at 0% equivalent to 4.66 J·kg^−1^·m^−1^. Where y = energy cost; x = gradient (%): y = 30.4x^4^ − 5.0975x^3^ + 46.3x^2^ + 17.696x + 4.66 (new energy cost equation).

**Figure 3 ijerph-17-09554-f003:**
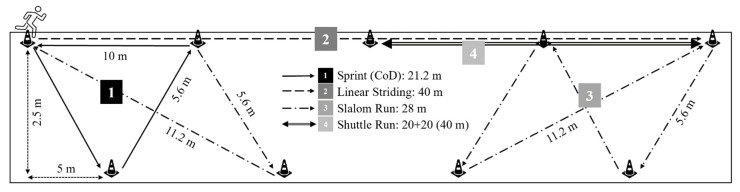
Soccer-specific intermittent exercise protocol.

**Figure 4 ijerph-17-09554-f004:**
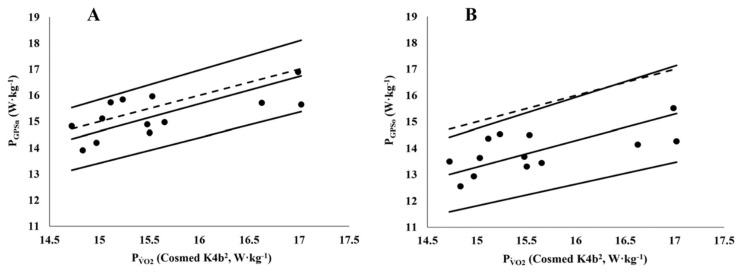
Ordinary least products regression of P_GPS_ values on $P_{\dot{V}O_{2}}$

**Figure 5 ijerph-17-09554-f005:**
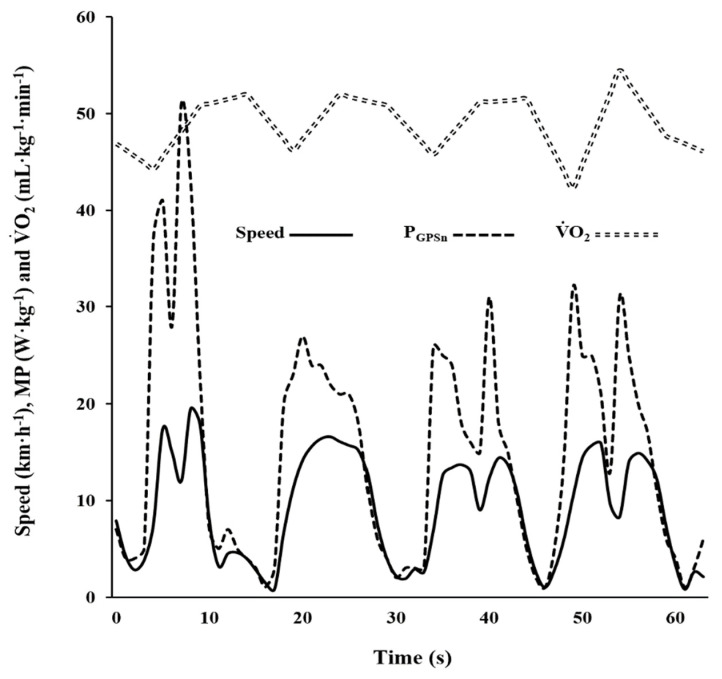
Oxygen uptake ($\dot{V}O_{2}$), speed and metabolic power estimated from locomotor demands (P_GPSn_) during one lap (1 min) of the soccer-specific circuit of one representative player.

**Figure 6 ijerph-17-09554-f006:**
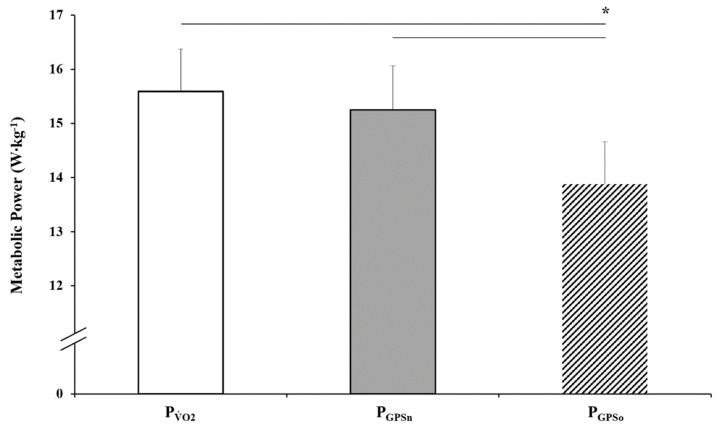
Metabolic power derived from direct measurement ($P_{\dot{V}O_{2}}$) and indirect assessment using two energy cost equations (P_GPSn_ and P_GPSo_). Statistical significance is denoted as “*”, *p* < 0.001, over MP conditions.

**Figure 7 ijerph-17-09554-f007:**
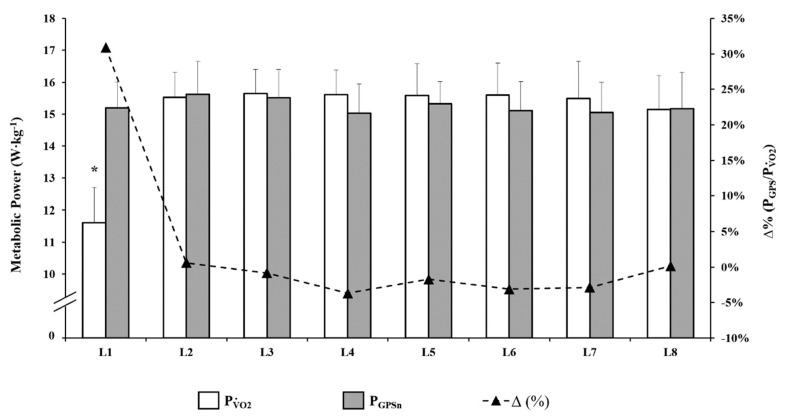
Metabolic power using either traditional calorimetry with oxygen uptake ($P_{\dot{V}O_{2}}$) or locomotor-related metabolic power (P_GPSn_) during the 8 laps of soccer-specific circuits (L1–L8). Statistical significance is denoted as “*”—*p* < 0.05.

**Table 1 ijerph-17-09554-t001:** Soccer-specific intermittent exercise protocol activities.

Activity		Distance (m)	Intensity (km·h^−1^/MP)	Time (s)
1	Sprint: triangle (change of direction > 60°)	21.2	Max (>26) *	~4–6″
2	Linear Striding	40	14.4 (23) *	~10″
3	Slalom Run	28	10.1 (20) *	~10″
4	Shuttle Run	20 + 20 (40)	14.4 (24) *	~10″

Note: * Exercise intensity expressed as MP (metabolic power (W·kg^−1^)).

**Table 2 ijerph-17-09554-t002:** Bioenergetics variables.

Bioenergetics Variables	Mean ± SD
$\dot{V}O_{2\max}$ (mL·kg^−1^·min^−1^)	55.7 ± 3.4
$\mathrm{Rest}\;\dot{V}O_{2}$ (mL·min^−1^)	266.0 ± 18.0
Steady-state running $\dot{V}O_{2}$ (L·min^−1^)	2.9 ± 0.3
Steady-state running $\dot{V}O_{2}$ (mL·kg^−1^·min^−1^)	40.8 ± 3.0
C_r_ (J·kg^−1^·m^−1^)	4.66 ± 0.4

**Table 3 ijerph-17-09554-t003:** Estimates of fixed and proportional bias from ordinary least products regression and their 95% CIs for P_GPSn_ and P_GPSo._

	Fixed Bias or Intercept	Proportional Bias or Slope
P_GPSn_ (Minetti et al., 2002 modified)	−0.803 (−8.393–6.788)	1.030 (0.531–1.528)
P_GPSo_ (Minetti et al., 2002)	−1.591 (−9.358–6.177)	0.992 (0.482–1.502)

**Table 4 ijerph-17-09554-t004:** Physiological and metabolic responses to the soccer-specific circuit.

Bioenergetic Variables	Mean ± SD
$\dot{V}O_{2\max}$ (mL·kg^−1^·min^−1^)	61.1 ± 4.3
Rest $\dot{V}O_{2}$ (mL·min^−1^)	268.0 ± 21.6
HR_max_ last 2 min (%)	94.0 ± 2.0
[La^−^]b (mmol·L^−1^)	7.2 ± 1.6
$\dot{V}O_{2}$ exercise (O_2_ debt included) [L·min^−1^]	3.4 ± 0.4
$\dot{V}O_{2}$ exercise (O_2_ debt included) [mL·kg·min^−1^]	44.5 ± 2.3
Energy cost (J·kg^−1^·m^−1^)	6.41 ± 0.31
$P_{\dot{V}O_{2}}$ (W·kg^−1^)	15.6 ± 0.8

**Table 5 ijerph-17-09554-t005:** Time (s) and Distance (m) were obtained during soccer-specific circuit performance in each speed and power categories.

Speed (*v*)/Power Categories	Distance (m)	Time (s)
*v* > 6 km·h^−1^	1060 ± 42	286 ± 11
*v* > 11 km·h^−1^	902 ± 65	220 ± 10
*v* > 16 km·h^−1^	277 ± 143	57 ± 29
*v* > 20 km·h^−1^	12 ± 14	2 ± 2
P_GPSn_ 0–10 W·kg^−1^	171 ± 35	210 ± 9
P_GPSn_ 10–20 W·kg^−1^	469 ± 33	123 ± 15
P_GPSn_ > 20 W·kg^−1^	531 ± 71	143 ± 13
P_GPSn_ 20–35 W·kg^−1^	373 ± 52	97 ± 11
P_GPSn_ > 55 W·kg^−1^	37 ± 12	11 ± 4

All data are expressed as mean ± SD.

**Table 6 ijerph-17-09554-t006:** Bioenergetic demands of the soccer-specific circuit.

External Load	Mean ± SD
Total distance (m)	1168 ± 53
P_GPSn_ (W·kg^−1^)	15.3 ± 0.8
High acceleration > 50% a_max_ (% time)	21 ± 3
High deceleration < −2 m·s^−2^ (% time)	17 ± 3
*v* > ES (% time/total time)	12 ± 6
Bouts·min^−1^ > 20 W·kg^−1^ (n)	4.7 ± 0.7
CoD > 30°·min^−1^ (n)	21 ± 2

Note: (*v*) speed; (ES) endurance speed; (CoD) change of direction.

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
