# Peer review of "The Validity of an Updated Metabolic Power Algorithm Based upon di Prampero’s Theoretical Model in Elite Soccer Players"

_ijerph, 2020, doi:10.3390/ijerph17249554_

Round 1

Reviewer 1 Report

Dear authors,

  1. your paper implicitly assumes an expert audience. It has, however, value for outside readers, who will skim this paper. Therefore I recommend replacing everywhere the mathematical term "linear running" by more everyday terminology, like "running at constant speed" as you do too late, in line 342.
    See for instance lines 19, 49, 50-51, 83, etc. 

    Also in view of a larger audience, I recommend replacing the term "steady-state" from automata theory, in line 124, into something which makes it comprehensible to me:  such as you do too late in this text in the lines 191-192.

  2. The lines 29-31 have a too-large font in the Abstract.

  3. The Section "2.5 Soccer-specific running" has a flaw: the pre- and post  blood lactate [La-]b test is said twice: lines 129-130 and lines 136-137 are almost similar. 

    Also is this section 2.5: it holds the same remark on the soccer boots at lines 138 - 140 is similar to your explanation in lines 169-170. 

    Is it a good idea to merge the  Section "2.5 Soccer-specific running" in the current  Section 2.7 ?

  4. In line 148, the word 'new' should be 'a new'. Is the wording "... impairs the calculation of MP." an error?  It is unclear to me how a new constant could impair computation of its polynomial.

  5. In line 151 you say '. . . it is determines during . . .' this seems in error. Do you mean here the past tense: 'detemined' ?

  6. The formul at line 216 should be in-line. It is a conversion, which - in my view - does not need a specific display. 
  7. In line 234 you forget to mention what the outcome of the normality test is.

  8. The Figures 6 and 7 are to be improved.
    In order to easily read the height of the bars I recommend to place the bars in Figure 6 closely together, near the vertical axis. Or if you do not like this: put a grid in the plot. This is easily done with a statistical/mathematical package Maple, by the grid parameter (Maplesoft.com).  

    The Figure 7 is to be improved. Currently it is poor.
    It should be a scatter- or point- or cloud-plot. One axis for the Pvo2 and the other axis for the GPSn. If also put the neutral line in the plot, for instance under the angle of 45 degrees, then the picture is improved specifically if you also put a grid in the plot. Again, this is easily done with a statistical/mathematical package Maple, by the grid parameter (Maplesoft.com).  

  9. I think that in the discussion some attention is needed for the polynomial modelling (Figure 2). Do the terms of the 4th order model get a physiological meaning? 

    From the mathematical viewpoint you could fit a N-1 order polynomial through N datapoints. But without any semantics the addended terms are meaningless. 

    You point in line 566 to enlargement of the analysis spectrum. Is this the same issue about the meaningless polynomial?

    The paper offers so much good work, shouldn't this be imbedded/attempted?

Author Response

Reviewer 1

Dear authors,

Your paper implicitly assumes an expert audience. It has, however, value for outside readers, who will skim this paper. Therefore I recommend replacing everywhere the mathematical term "linear running" by more everyday terminology, like "running at constant speed" as you do too late, in line 342. See for instance lines 19, 49, 50-51, 83, etc. 

R: Thanks for your comment, we have changed the term linear-running to constant-speed linear-running to follow the term used in line 342 and help the readers.

Also in view of a larger audience, I recommend replacing the term "steady-state" from automata theory, in line 124, into something which makes it comprehensible to me:  such as you do too late in this text in the lines 191-192.

R: Thanks for your comment. In line 124 we are describing the method used for calculating Cr discussing in small physiological details to provide a better understanding of steady-state intensities and the energy demand required.

The lines 29-31 have a too-large font in the Abstract.

R: Thanks for your comment. The font has been changed to the same throughout the abstract.

The Section "2.5 Soccer-specific running" has a flaw: the pre- and post blood lactate [La-]b test is said twice: lines 129-130 and lines 136-137 are almost similar. 

R: Thanks for your comment. We have taken out the 2nd flaw in lines 129-130 to ensure no repetition is made.

Also is this section 2.5: it holds the same remark on the soccer boots at lines 138 - 140 is similar to your explanation in lines 169-170. 

R: Thanks for your comment. We deleted the first sentence in 2.7. (line 169) to ensure we do not have the same explanation in both sections.

Is it a good idea to merge the Section "2.5 Soccer-specific running" in the current Section 2.7?

R: Thanks for your comment. To better clarify each step we described the protocol to avoid a possible confusion. We had these sections together initially, but it was extremely confusing and unclear.

In line 148, the word 'new' should be 'a new'. Is the wording "... impairs the calculation of MP." an error?  It is unclear to me how a new constant could impair computation of its polynomial.

R: Thanks for your comment. We changed accordingly to your suggestion

Regarding the term “impairs the calculation of MP”; the reason: it was negatively affects the MP derived, because the energy cost from the polynomial equation in the original approach of Minetti et al. 2002 would be underestimated due to the slope % (Fig 2; example C of -60% < 50%).

In line 151 you say '. . . it is determines during . . .' this seems in error. Do you mean here the past tense: 'determined'?

R: Thanks for your comment. It is determinates has been changed to “is determined”.

The formula at line 216 should be in-line. It is a conversion, which - in my view - does not need a specific display. 

R: Thanks for your comment. We have now put the equation in-line of your suggestion in a correct font.

In line 234 you forget to mention what the outcome of the normality test is.

R: Thanks for your comment. We included this information “normal distribution” in Statistical Section. Besides, to better clarify the results, We included the ANOVA with repeated measures (Bonferroni Post-Hoc) and Fisher value.

The Figures 6 and 7 are to be improved.
In order to easily read the height of the bars I recommend to place the bars in Figure 6 closely together, near the vertical axis. Or if you do not like this: put a grid in the plot. This is easily done with a statistical/mathematical package Maple, by the grid parameter (Maplesoft.com).  

R: Thanks for the note. Figure 6 with this arrangement (y axis) highlights the differences between the methods in a better way.

The Figure 7 is to be improved. Currently it is poor.
It should be a scatter- or point- or cloud-plot. One axis for the Pvo2 and the other axis for the GPSn. If also put the neutral line in the plot, for instance under the angle of 45 degrees, then the picture is improved specifically if you also put a grid in the plot. Again, this is easily done with a statistical/mathematical package Maple, by the grid parameter (Maplesoft.com).  

R: Thanks for the note. Figure 7 has been modified by changing the scale of the y axis to better represent the percentage differences between PV̇O2 and PGPS (second vertical axis). Thus also adding the statistical significance present in the first minute of the circuit (L1), which highlights the greater delta (%), due to the onset of oxygen consumption lagging behind the kinematic parameters (indirect energy calculation).

I think that in the discussion some attention is needed for the polynomial modelling (Figure 2). Do the terms of the 4th order model get a physiological meaning?

R: Thanks for your comments. Physiologically the 4th order polynomial modelling is just useful to better describe the limitation established from the Minetti et al. 2002 equation. Therefore, this was not discussed in the discussion, but a detailed description was provided within the methodology (2.6).

From the mathematical viewpoint, you could fit a N-1 order polynomial through N datapoints. But without any semantics the addended terms are meaningless. 
R: Thank you for this clarification which highlights the observation and knowledge of the mathematical approach to the problem. The lowering of degree (N-1, 5th > 4th) towards a polynomial which better fit any datapoints and describes our physiological goal, is an “artificial reduction” that is not based on directly calculated experimental points (as done by Minetti et al. 2002). Its de facto meaning is the possibility of mathematically intervening on the energy cost expressed by the inflection point of the equation (> -45% slope) which would involve a counter-intuitive error compared to the energy cost of high decelerations.

You point in line 566 to enlargement of the analysis spectrum. Is this the same issue about the meaningless polynomial?
R: Thanks for your comment. The clarity of the sentence has been altered to highlight that the issue is related to player-specific analysis for soccer.

The paper offers so much good work, shouldn't this be imbedded/attempted?

R: Thanks for your comments. The main aim of the paper was: 

1) to determine the energy cost of running on grass (Cr) in ecological conditions on elite soccer players and 2) to validate an updated MP with a new equation using a soccer-specific exercise protocol.

Therefore, we provide information which is related to the above aims, without going in-depth regarding other aspects which are purely “mathematical” to take into account that the paper is not only implicitly assuming an expert audience. We are also aware that the paper is long and “quite complex” and ensure that relevant information regarding the main physiological aims are provided without focusing in-depth on the other aspects.

Reviewer 2 Report

What is the novelty of the paper. It is not clear the idea of the paper. How many samples were included in the analysis? It is not clear how authors control the control variables. What is the practical implication of this paper? 

Author Response

Reviewer 2

What is the novelty of the paper - It is not clear the idea of the paper.

R: Thanks for your question/comment. This manuscript is based on other research in the literature looking at the validation of MP while taking into account the limitations previously observed. Previous metabolic demands imposed by soccer match-play and training estimates are calculated from energy cost paradigms derived from laboratory models of constant-speed linear-running, that do not reflect the totality of soccer related actions.

A more specific intermittent “protocol” was utilized within this research to determine a more “accurate” C using professional soccer players. The main aim of the study is to demonstrate that the exercise undertaken influences the MP largely compared to the biomechanical theory and the equation of energy cost.

How many samples were included in the analysis? It is not clear how authors control the control variables.

R: Thanks for your question. As discussed within the methodology, seventeen first-team professional male soccer players competing in the Italian Serie A [mean ± SD: age 24.0 ± 2.9 yrs, stature 175.1 ± 4.9 cm, body mass 75.9 ± 5.2 kg and V̇O2max 55.7 ± 3.4 mL∙min-1∙kg-1] were recruited for the first part of this study to determine constant-speed linear-running energy cost (Cr). Subsequently, thirteen of these players [age (mean ± SD) 22.1 ± 5.9 yrs, height 181.8 ± 5.4 cm and body mass 76.5 ± 6.2 kg] then performed a soccer-specific intermittent exercise protocol to determine soccer energy cost (C). See 2.1.

What is the practical implication of this paper? 

R: Thanks for your question. The practical implication of the paper is to provide a better explanation of the concept of MP applied to team-sports. Focusing on the protocol used in this study, Now We are able to provide a clearer understanding of the “importance” of MP within soccer compared to other sports, which may not be suitable to this concept derived from the kinematics values. Anyway this investigation provide a better representation of intermittent exercise/game demands thus typical in soccer game. Findings are useful to construct a specific “training” routine knowing the limitation of any intermittent nature bouts.

Reviewer 3 Report

The present study aimed to determine the energy cost of elite professional soccer players and propose an updated metabolic power algorithm using a soccer-specific performance model. 

Congrats to the authors for this wonderful study. I can see your hard works on the present manuscript. Some questions or suggestions are provided to the authors.

Introduction:

The introduction of this study was well performed and provided its necessity.

Materials and Methods:

  1. Due to an updated algorithm proposed for publication, is the sample size enough for your analysis?
  2. A small sample may limit the statistical methods. For example, the Pearson correlation should be used for a large sample size (n > 30). I suggest the authors recruit more participants into your study.
  3. The authors mentioned that the grass soccer field was used for the study protocol, with appropriate soccer footwear. However, even based on the UEFA standard field, there are remain differences among the stadiums. For example, the grass itself, soil moisture, and the footwear the participants wore. These factors are also influencing the "running economy" I suggest the authors provide related information or discuss this issue.

Results:

  1. The r-values seem minor to update the existing method, although we didn't know it reliable yet.

Discussion:

  1. some sentences and paragraphs are wordy and hard to be read.

Author Response

Reviewer 3

The present study aimed to determine the energy cost of elite professional soccer players and propose an updated metabolic power algorithm using a soccer-specific performance model.

Congrats to the authors for this wonderful study. I can see your hard works on the present manuscript. Some questions or suggestions are provided to the authors.

R: Thank you for your kind comments and taking the time to review this manuscript

Introduction:

The introduction of this study was well performed and provided its necessity.

R: Thanks for your comments.

Materials and Methods:

  1. Due to an updated algorithm proposed for publication, is the sample size enough for your analysis?

R: Thanks for your question. We included this point in the limitation’s study.

  1. A small sample may limit the statistical methods. For example, the Pearson correlation should be used for a large sample size (n > 30). I suggest the authors recruit more participants into your study.

R: Thanks for your comment. As mentioned above, the statistical limitation is part of using an elite group of participants. Few studies investigated elite soccer players and this study provides novel information especially regarding the energy cost for this specific population. Anyway, the data follow a normal distribution as included in the statistical section.

  1. The authors mentioned that the grass soccer field was used for the study protocol, with appropriate soccer footwear. However, even based on the UEFA standard field, there are remain differences among the stadiums. For example, the grass itself, soil moisture, and the footwear the participants wore. These factors are also influencing the "running economy" I suggest the authors provide related information or discuss this issue.

R: Thanks for your comments. This is why we clearly specify the place (ground) that the testing took place, as we are aware about the differences in the impact of different terrain. For this reason we mention/discuss the previous studies of Pinnington & Dawson, and Sassi et al. which provide detailed information regarding the influence of the ground. This point it was discussed within the manuscript.

Results:

  1. The r-values seem minor to update the existing method, although we didn't know it reliable yet.

R: Thanks for your comment. Among the aims of the study there is the correlation evaluation, but there is no acknowledged comparison between the methodology object of study and the assumed golden standard (i.e., PV̇O2). The reader would expect to find the proper statistical technique to investigate the device’s effectiveness, i.e., precision (or validity): the Ludbrook’s method [Comparing methods of measurements; Statistical techniques for comparing measurers and methods of measurement: a critical review; Linear regression analysis for comparing two measurers or methods of measurement: but which regression?]. Differently, the authors studied the device’s effectiveness by un-appropriately focusing on the correlation.

Discussion:

  1. some sentences and paragraphs are wordy and hard to be read.

R: Thanks for your comments. Several changing was made to better clarify the main document.

Round 2

Reviewer 2 Report

Accept

Reviewer 3 Report

The revision from the authors has satistified the reviewer. Congrats.